# Awareness and Knowledge of Rare Diseases in German Dentists, Dental Specialists and Oral and Maxillofacial Surgeons: A Country-Wide Survey

**DOI:** 10.3390/medicina58081114

**Published:** 2022-08-17

**Authors:** Korbinian Benz, Ronny Trapp, Maximilian Voss, Marcel Hanisch, Urban Geisthoff, Jochen Jackowski

**Affiliations:** 1Department of Oral Surgery and Policlinical Ambulance, Faculty of Health, Witten/Herdecke University, Alfred-Herrhausen-Str. 45, 58448 Witten, Germany; 2Department of Cranio-Maxillofacial Surgery, University Hospital Muenster, Albert-Schweizer-Campus 1, 48149 Muenster, Germany; 3Ear, Nose and Throat Clinic, University Hospital Giessen and Marburg GmbH, Marburg Site, Baldingerstraße, 35043 Marburg, Germany

**Keywords:** rare diseases, public health, medical education, information needs, online survey

## Abstract

*Background and objectives*: Rare diseases affect an estimated four million patients in Germany. Approximately 15% of the approximately 6000 to 8000 rare diseases known globally show manifestations in the dental, oral and maxillofacial regions. The present survey evaluated the knowledge and management of rare diseases and their orofacial alterations by dentists, dental specialists and oral and maxillofacial surgeons and dentists working at university hospitals for dentistry and/or oral and maxillofacial surgery. *Materials and Methods:* The study was designed as an anonymous cross-sectional study. Two anonymous online surveys were performed in all dentists in Germany using the open-source survey software limesurvey. The study cohorts were divided into dentists, dental specialists and oral and maxillofacial surgeons in practice, and dentists who worked in university dental and oral and maxillofacial surgery centers. The survey was performed between 1 October 2020 and 31 March 2021. *Results:* A total of 309 dentists and oral and maxillofacial surgeons in private practice and 18 dentists or oral and maxillofacial surgeons working at universities participated. A total of 209 (86.7%) study participants working in private practice indicated that the topic of rare diseases should be considered clinically relevant. University participants indicated that there was a lecture on rare diseases in only 7 (63.6%) cases. Only 2 (13.3%) participants reported active research on the topic in their department. *Conclusions:* The current knowledge on rare diseases is inadequate in suitable screening and therapy. Most of the participants believed that knowledge of rare diseases was very important for daily dental practice. The self-estimations showed that all of the participants estimated their knowledge as very good or inadequate, with a tendency in the direction of inadequate knowledge.

## 1. Introduction

Rare diseases (RD) affect a small number of patients who have limited or no therapy options. The European Union (EU) definition of an RD is a disease that affects fewer than 5 in 10,000 people [1]. Between 7000 and 8000 of the approximately 30,000 known diseases are classified as rare. However, the number of RDs is not small compared to the total number of all diseases, but the individual diseases are nevertheless responsible for few or the fewest number of cases. RD poses a special challenge to everyone in the health care sector due to the smaller numbers of patients with chronic/progredient progress; disabilities; massive organ damage; and, in some cases, a high level of mortality. Approximately 80% of RDs are hereditary and manifest in newborns and infants, but other RDs only manifest in adulthood. The classification of “rare” is subject to temporal and regional aspects. Morbus Behçet has a prevalence of 0.6 cases/100,000 inhabitants in Germany, but this prevalence increases to 1–8 cases/10,000 inhabitants in the Mediterranean area, the Middle East and Japan. There is a manifestation of orofacial changes in approximately 15% of all RDs [2]. Between 6–8% of the EU population requires multiprofessional and long-term medical care due to an RD. Approximately four million people are suffering from an RD in Germany [3]. This prevalence highlights the significance of RDs for healthcare workers in private practice and hospitals [4]. It is impossible to provide precise knowledge about any RD because there are so many of them. Therefore, collaboration across disciplines is crucial between the fields of medicine and dentistry [5]. For instance, oral signs of (gastrointestinal) Crohn’s disease, such as cheilitis and ulcerations of the oral mucosa, are possible [6]. Ectodermal dysplasia is characterized by a symptom triad that includes an anomaly, such as a conical tooth form or missing teeth [7]. When combined with skeletal changes, dentogenic abscesses on caries-free teeth may be an X-linked hypophosphataemia (phosphate diabetes) sign [8]. Gorlin-Goltz syndrome may be indicated by several basal cell carcinomas in addition to recurrent odontogenic keratocysts [9]. According to the European Medicines Agency (EMA), only the most prevalent (fewer than 1000) of the up to 8000 RDs appear to benefit from sparce medical and scientific knowledge, despite the fact that there have been a rising number of publications on RDs [10].

The patients and their relatives are subjected to increased psychological burdens due to unclear diagnoses, a lack of available therapies, and the restricted support of the existing healthcare facilities. The complex nature of RDs makes special knowledge and interdisciplinary expertise necessary. More physicians or hospitals with the required medical knowledge, diagnostic means and organizational structures are necessary to ensure a correct diagnosis and adequate therapy in the scope of patient-centered care.

Few studies examined the knowledge and awareness of physicians and dentists on RDs.

The present study internally developed a questionnaire and performed a national evaluation on how German dentists/dental specialists/oral and maxillofacial surgeons (OMFS) in practice deal with patients suffering from RD and their needs for knowledge of RDs and further training in this field. The survey was also offered to German schools of dentistry and university departments of oral and maxillofacial surgery, with an additional questionnaire about tutoring, scientific activities, and further training in the RD field.

## 2. Materials and Methods

### 2.1. Study Design

The study was an anonymous cross-sectional study. Two anonymous online surveys were offered to all the dentists in Germany using open-source survey software (https://www.limesurvey.org (accessed on 30 September 2020, Hamburg, Germany). The study cohorts were divided into dentists, dental specialists and OMFS in practice, and colleagues who worked at university dental and/or oral and maxillofacial surgery centers.

The survey was conducted between 1 October 2020 and 31 March 2021.

### 2.2. The Questionnaires

#### 2.2.1. Questionnaire for Dentists and Specialists for Oral and Maxillofacial Surgery in Practice

The questionnaire for the 1st study cohort was divided into 5 sections and included a total of 37 questions. Twenty-seven of these questions provided a simple selection of the provided possible answers. Eight questions were multiple choice with an additional option of a free text answer, and 2 questions able to be answered in free text form.

The first section, “Questions concerning the Person, the Work Environment and the Qualification”, collected data using 12 questions about the participant, including the work environment, gender, age, health insurance license, single or multiple dentist(s) practice, professional experience, specialization, telematic infrastructure and participation in the quality circle.

The second section, “Professional Cooperation in the Field of Rare Diseases with Orofacial Participation”, collected data using 8 questions on cooperation with specialized colleagues, the number and reason for contact with patients suffering from an RD, the necessary referral of patients, and external consultations with colleagues on patients suffering from an RD.

The third section, “Use of a Register for Rare Diseases with Orofacial Manifestation”, collected data from 5 questions on the assistance and benefits provided by an RD register, the sources of information in current use, RD databases, and further training courses that could be used to improve knowledge.

The fourth section, “Professional Exchange in the Field of Rare Diseases with Orofacial Participation”, collected data from 10 questions on the wishes and disadvantages that result from cooperating with colleagues who specialize in the treatment of RD; the remuneration paid for the treatment of patients suffering from RD; the exchange using visual media, e.g., a video conference; further training and knowledge improvement on the basis of a quality circle; and the knowledge that centers have for RD.

The fifth section collected data from 2 questions on the estimation of the clinical relevance of the subject of RD.

#### 2.2.2. Questionnaire for Colleagues at University Dental/Oral and Maxillofacial Surgery Centers

The questionnaire for the 2nd study cohort was divided into 4 sections and included a total of 21 questions. Twelve of these questions allowed for a simple selection of provided possible answers. Six questions were multiple choice with an additional option of a free text answer, and 3 questions allowed answering in free text form.

The first section, “Department”, collected data from 10 questions on the work environment, the function of the activities, the size of the department, the number of colleagues treating patients suffering from RDs and their further training, and the existence and scope of surgery for RDs.

The second section, “Patients”, collected data from 3 questions on the number of treated patients, who referred the patients to surgery, and the orofacial symptoms of the oral and maxillofacial surgery that were treated.

The third section, “Research in the Rare Diseases Field”, collected data from 3 questions on research work within the department, the number of scientific publications and the lecturing activities of the department.

The fourth section, “Lecturing”, collected data from 5 questions on lectures on RDs and the possibility for student attendance.

### 2.3. Ethics Committee Vote

The ethics committee at the University of Witten/Herdecke provided notification of its positive ethics committee vote on 11 March 2020, assigning the reference number 20/2020.

### 2.4. Participants

The questionnaires or link to both of the surveys were distributed via various media, with a corresponding explanation and invitation to participate. The information was published in “*Zahnärztliche Mitteilungen*” magazine (zm 110, issue 19, 1 October 2020), the associated online portal with the newsletter (https://www.zm-online.de 1 October 2020/zm-online newsletter 7 October 2020) and “*Oralchirurgie Journal*” magazine (issue 1/2021). A press release was also issued in cooperation with the German Federal Association of Dentists on 1 October 2020, and this release was subsequently published in the “Bayerisches Zahnärzteblatt” in Bavaria, the “Berliner Zahnärzte Meinungsblatt”, and the “Zahnärzteblatt Rheinland-Pfalz” in the Rhineland-Palatinate.

### 2.5. Statistical Analysis

Statistical analyses were performed using Stata/IC 16.1 for Unix StataCorp (4905 Lakeway Drive College Station, College Station, TX, USA).

## 3. Results

### 3.1. Replies of Dentists/Dental Specialists and Oral and Maxillofacial Surgeons in Practice

This section summarizes the results of the questionnaire about RDs for dentists in practice. A description of how frequently the various answers were provided is given for each question.

There are currently a total of 97,372 dentists in Germany, of whom 72,592 are actively practicing. Of these, 50,022 are in private practice and 22,570 are employed in practices, medical care centers or health care facilities. Information from 309 participants who were asked and answered at least some of the questions was recorded. Some of the participants only provided information in the first section, “Questions concerning the Person, the Work Environment and the Qualification”, and 30 of the 309 participants did not provide any information in Sections II–IV. Table 1 shows the frequency of the various answers provided by the 309 participants, e.g., the number and distribution of the salaried employees and the self-employed. Because most of the questions were relatively long, the specific wording is provided underneath the tables for reasons of clarity. The tables merely include the numbers of the questions.

Information is also provided on the number of asked participants who answered Sections II–IV, i.e., the number and percentage of participants who were in salaried employment and provided information in Sections II–IV (Information in Sections II–IV (yes/no) column). The table also includes a column to the far right that describes the distribution among the reply categories by participants who provided information in Sections II–IV (i.e., the number and distribution of the salaried employees and the self-employed who answered Sections II–IV). A total of 279 of the 309 participants answered all of the questions in the questionnaire. Of these participants, 146 (52.3%) were female, 130 (46.6%) were male, and 3 (1.1%) were of a diverse gender. A total of 124 (44.4%) stated that they were working in rural areas, and 155 (55.6%) stated that they were working in urban areas. A total of 118 (42.3%) worked in a single practice, and 161 (57.7%) worked with at least one other colleague. For professional experience, 91 (32.6%) stated less than 11 years, 58 (20.8%) stated between 11 and 20 years, 71 (25.4%) stated between 21 and 30 years, 47 (16.8%) stated between 31 and 40 years, and 12 (4.3%) stated that they had more than 40 years of professional experience. A total of 233 (83.5%) of the participants worked as dentists without further training as a dental specialist. Thirty (10.8%) participants stated that they were qualified oral surgeons, 8 (2.9%) were qualified orthodontists, 7 (2.5%) were qualified periodontists and 1 (0.4%) was a dental specialist for the public health service. The use of a telematic infrastructure was negated in 81 (29.0%) questionnaires, and 198 (71.0%) participants confirmed having initiated it. A total of 122 (61.6%) participants stated that they used it “frequently”, and 76 (38.4%) stated that they tended to use it “seldomly”. Thirteen (4.7%) participants deemed participation in quality circles as “pointless”, 111 (39.8%) had “never” participated in one, 105 (37.6%) participated in one “occasionally” and 50 (17.9%) dentists described their participation as “regular”.

Age is described as mean values with standard deviation and medians with minimum and maximum values. The mean age of the participants was 48.0 years old (Table 2).

A total of 61.4% (*n* = 62) of dentists who worked in a practice with one or more colleagues rarely (“seldom”) consulted each other when a patient suffering from an RD appeared. A total of 31.7% (*n* = 32) of the participants reported that consultations occurred “regularly”, and 6.9% (*n* = 7) reported “never”.

Forty (39.6%) participants stated that the reason for consultation was that an RD had already been diagnosed and necessitated co-treatment. Eighty-three (82.2%) of the consults were for obtaining a second opinion, and 15 (14.9%) participants stated that it was requested by the patients. Eight (7.9%) of self-employed participants stated that they had “never” consulted with a colleague, and this rate was “seldom” in 60 (59.4%) of the cases and “regular” in 33 (32.7%) of the cases. The reason for consultation was the RD for 45 (44.6%) dentists, to obtain a second opinion for 84 (83.2%) participants and a patient request for 15 (14.9%) participants. The exchange with external dentists, dental specialists or OMFS “never” occurred in 14 (7.9%) cases, “seldom” occurred in 86 (48.3%) cases and “regularly” occurred in 78 (43.8%) cases. A referral occurred in 95 (53.7%) cases with a frequency of 0–25%, in 11 (6.2%) cases with a frequency of 26–50%, in 20 (11.3%) cases with a frequency of 51–75% and in 51 (28.8%) cases with a frequency of 76–100%. Forty (22.7) of the dentists in practice stated that they never (0%) referred patients to a university dental hospital or clinic for oral and maxillofacial surgery, and the “max. 5%” in 81 (46.0%) cases, “max. 10% in 18 (10.2%) cases and “more than 10%” in 37 (21.0%) of the cases. A referral to a specialist occurred in 53.7% of the cases, and almost every fifth (22.7%) patient with an RD was referred to a university hospital (Table 3).

A total of 236 participants believed that the creation and provision of a register would be useful for diagnostics and therapy (87.7%) and communication with medical colleagues (88.1%). Five (1.9%) participants stated that they were unable to discern any benefits from it, and 28 (10.4%) and 27 (10.0%) were uncertain. For communication with medical colleagues, 237 (88.1%) participants indicated that this type of register would be beneficial, 5 (1.9%) participants indicated it would not be beneficial, and 28 (10.0%) were uncertain.

A total of 219 (80.8%) of the dentists in practice otherwise referred to medical reference books, and 123 (45.4%) consulted Medline. In contrast, publicly accessible databases played an insignificant role in the past: 220 (81.2%) participants stated that they had not used any of the stated sources. Orphanet was the chosen research media for 27 (10.0%) participants. A total of 163 (60.1%) dentists stated that congresses and symposiums were an appropriate medium for the imparting of knowledge in the RD field, and 115 (42.4%) indicated that interdisciplinary quality circles were helpful in this connection. A total of 149 (55.0%) participants deemed advanced telemedical training to be a suitable measure, and 23 (8.5%) participants referred to “other advanced training” (Table 4).

A total of 206 (79.5%) participants would like stronger cooperation with their colleagues, and 237 (91.5%) did not have any fears that they would “lose” the patient as a result of the referral. A total of 236 (91.1%) of the dentists in practice also indicated that a possible loss of competence was not critical. A total of 106 (41.1%) participants believed that the creation of remuneration incentives was necessary, with 82 (31.8%) disagreeing and 70 (27.1%) being uncertain. A total of 106 (41.1%) participants believed that the telematic infrastructure was appropriate for making a diagnosis and treating patients suffering from an RD easier, and 189 (73.0%) would avail themselves of telemedical care offered by a university dental hospital. A total of 155 (60.1%) of the participants believed that a regular quality circle would be a suitable measure for intensifying their knowledge in this field, and 33 (12.8%) were negatively disposed of this or were uncertain (70/27.1%). Of the participants who agreed, 63 (28.0%) wanted this to be on the basis of “numerous meetings”, with 205 (91.1%) wanting this to be on the basis of a “presentation of individual cases”. Fifty-four (20.8%) participants were aware of the existence of centers for RD, and 207 (79.2%) were not aware. Eighteen (33.3%) participants who stated that they were aware of the existence of such centers had already cooperated with one. A total of 209 (86.7%) of the participants stated that the subject “rare diseases” was clinically relevant, 13 (5.4%) replied “no”, and 19 (7.9%) had “no opinion” (Table 5).

### 3.2. Replies from Colleagues at University Centers for Dentistry and Oral and Maxillofacial Surgery

It was possible to analyze a total of 18 questionnaires, and the presentation of the results conformed with the presentation of dentists in practice. It was possible to analyze six different faculties from all of the faculties of dental medicine, numbering 29 in total. Table 6 presents the participating universities or their participants with the qualification/position. One conspicuous aspect is that only 2 (11.1%) participating university hospitals answered questions on a separate surgery for patients suffering from an RD. Special training for medical personnel was stated in 4 (22.2%) reports, and no special training was reported by 14 (77.8%) participants. Further training should be in the form of regular further training in 2 (50.0%) cases, and in the form of other “further training” in 3 (75.0%) cases, with 3 (75.0%) reports stating that this was in the form of collecting “internal clinical experience”.

Eight (50%) participants stated that they treat more than 20 patients per calendar year, and most (75.0%) of these patients were referred by external practices. Intraoral and extraoral symptoms were treated to the same extent (50%/50%) (Table 7).

The question of whether RD research was performed in the department was answered “no” in 86.7% of the cases. Participants who answered “yes” had a publication rate of “0” in one case and “20” in the other case in the past three years, with the departments concerned being represented at national/international congresses or not, respectively (Table 8).

The offer of a lecture on the subject of RD in connection with dental, oral and orthodontic diseases was confirmed by 36.4%. A total of 54.5% did not know whether other disciplines provided lectures in this specialization. A total of 45.5% stated that students were able to sit in on the treatment of patients with an RD in the form of sitting in/clinical traineeships (40.0%) or internships (60.0%) Table 9.

## 4. Discussion

Only one study of the status of knowledge on RD was performed in a small cohort of dentists, specialized dentists and OMFS in Germany. Most internationally published studies were performed on physicians or medical students. Therefore, this study identified potential knowledge gaps in RD and possible areas for improvement in the specialized basic and further training field. The negative self-assessment of the participants on the state of their knowledge of RD was especially because RDs are not given sufficient consideration in the curricula at universities. This insufficiency also has the consequence that most of the participants did not feel able to care for this group of patients [11,12,13]. In addition to the deficient medical training on the subject of RD, a large number of future physicians are not aware of the existence of Orphanet, which is a European register that provides information on this subject [13]. Because centers for the treatment of RD are not adequately known among colleagues, adequate caring for patients suffering from RD cannot be ensured to an adequate degree. In this first country-wide survey of Germany, only two university departments were actively conducting research on this subject. Zero or 20 scientific studies were published on the subject of RD during the past three years. However, this group of patients poses a special challenge during treatment, and RDs are a severe public health problem [12]. Most participants also showed interest in expanding their knowledge of RDs, including postgraduate education [13]. A high number of referrals of patients to specialists may directly correlate with a lack of knowledge as a result of inadequate basic and further medical training. The few consultations with colleagues in connection with RD suggests a lack of awareness on this subject. Because a large number of physicians are aware of their lack of knowledge, and there is a high level of readiness to undergo further training and a desire for stronger cooperation with colleagues in the future, there would certainly be an awareness and will to diagnose an RD at an early stage.

The knowledge and general awareness of RD was studied among family physicians in Bulgaria in 2011 [14]. Due to their proximity to the population, general practitioners were the first persons to come in contact with “unusual” patients who may suffer from RD. From a sample of 2042 general practitioners, 1002 consented to a telephone interview. The mean period of service was 18.54 ± 8.51 years for male physicians and 19.63 ± 7.81 years for female physicians (*p* < 0.05). Seventeen questions were formulated in five areas to evaluate knowledge and awareness of the complex RD theme. The replies to these subject blocks resulted in the determination of a low level of general knowledge on RD and a poor awareness of RD. Only one-fifth of the participants (*n* = 198, 19.82%) knew the exact definition of an RD in Europe. In view of the awareness that the family physicians had regarding the proportion of people with an RD when compared to the total population of a country (6–8%), only 69 of the participants (6.90%) knew the correct answer. More than half of them (*n* = 606, 60.50%) significantly underestimated the relevance for the social community and public health. The other family physicians (*n* = 327, 32.60%) were unable to answer this question. When combining these aspects (the definition of an RD and the distribution among the population in a country), only 23 (*n* = 2.3%) knew the correct answers. General practitioners were unable to provide their patients who were suffering from RD with information about their RD that had adequate quality and without delay. Particularly alarming was the underestimation of the prevalence of RD by the general practitioners. The possibility of a medical genetic consultation was not used to the required extent. A recommendation was made for a campaign that aimed to increase the awareness of RD by general practitioners, with the aspects of prevention, diagnostics, treatment and rehabilitation. The authors of this study mentioned that facultative courses for medical students and postgraduate further training courses for physicians could expand the knowledge of the frequency of RD and present an adequate concept for the handling of “difficult” patients.

A Belgian study from 2019 [11] surveyed Belgian general practitioners, pediatricians, neurologists, and endocrinologists to determine their awareness and knowledge of RDs. Twenty-five multiple-choice questions were drafted and answered online. The survey was performed on the basis of interviews with 9 experts for RD. It was anonymous, and a total of 295 physicians (*n* = 295) participated. Thirty-nine percent of the participating physicians were general practitioners, 32% were pediatricians, 9% were “pediatric specialists” and 16% were “adult specialists”. The questionnaire was comprised of 25 multiple choice questions in five sections. Forty-four percent of the participants (*n* = 129) did not work in a hospital, 31% (*n* = 90) worked in a university hospital, 22% (*n* = 65) worked in a non-university hospital and 3% (*n* = 10) worked in an “other” category. For self-estimation, most of the general practitioners (*n* = 113) estimated their knowledge of RD as being “reduced” (44%) or “poor” (42%). For pediatricians (*n* = 94), 36% estimated their knowledge in this regard as being “reduced” or “average”. The pediatric specialists (*n* = 27) estimated their knowledge best and estimated their knowledge as “good” (44%). Most of the family physicians also stated that they did not have any awareness of RD (32%) or that their knowledge was inadequate (39%). Exactly one-quarter (25%) of the pediatricians stated that they had a “very good awareness” of RD, and an additional 28% stated that they had a moderate awareness, with 29% only having a poor awareness. Thirty-two percent of the general practitioners, 5% of the pediatricians, 11% of the pediatric specialists and 9% of the other medical specialists answered the question, “Do you have an awareness of rare diseases?” with “absolutely not”. Another aspect of the inadequate knowledge of RD may be inadequate medical training. Only 1% of the family physicians, 2% of the pediatricians, 4% of the pediatric specialists and 8% of the other medical specialists reported their medical training in the diagnosis of RD as “very helpful”. Most of the family physicians (57%) described this part of their training as “moderately useful”, 39% of the pediatricians described it as “inadequately useful”, and 48% of the pediatric specialists and 40% of the other medical specialists characterized it as “adequately useful”. The “Orphanet” online platform was the most well-known source of information. Twenty-two percent of the family physicians, 85% of the pediatricians, 89% of the pediatric specialists and 75% of the other medical specialists stated that they were at least aware of Orphanet as a source of information.

Another study of dentists, dental specialists and OMFS was performed in the Westphalia-Lippe district of Germany, which is the location of the University Dental Hospitals in Münster and Witten/Herdecke. It was in the form of an anonymous cross-sectional study and was performed in 2019 [15]. The structure guidelines were based on the previously presented Belgian study [11]. The questionnaire for this study was broken down into five sections and was comprised of 24 questions. A total of 267 dentists (*n* = 267) completed the questionnaire fully. A total of 171 of the 267 participants were general dentists, 68 were dental specialists (239 outside universities) and 28 were dentists at a university hospital. The dental specialists included 28 oral surgeons, 9 OMFS and 19 orthodontists. Therefore, it was possible to analyze this survey on the basis of a differentiation between general dentists, dental specialists and dentists at university hospitals. The aim of this study was to disclose potential gaps in the knowledge of RD and develop possible improvements for a lack of basic and advanced dental studies. A total of 64.3% (*n* = 18) of the dentists at university hospitals, 57.4% (*n* = 39) of the specialists and 48% (*n* = 82) of the general dentists estimated that an RD was a disease that affected no more than five of 10,000 people in the EU. A total of 32.1% (*n* = 9) of the dentists at a university hospital, 35.3% (*n* = 24) dental specialists and 43.3% (*n* = 74) of the general dentists guessed that no more than five of 250,000 people were affected in the EU. A total of 39.3% (*n* = 11) of the dentists at a university hospital were aware that 15% of RDs manifested in the craniomaxillofacial area. This response was differentiated from the replies of the other two groups (*p* = 0.029). The correct answer was given by 17.6% (*n* = 12) of the dental specialists and 15.8% (*n* = 27) of the general dentists. There was a statistical difference (*p* = 0.012) between the three groups “general dentists”, “specialist dentists” and dentists working at a university, for the question on the RDs that the participants were aware of. Dentists working at a university were aware of more RDs, as was the case with dental specialists (*p* = 0.007) and general dentists (*p* = 0.004). A total of 75.0% (*n* = 21) were aware of eight or more RDs, and none of the dentists who worked at a university hospital were aware of fewer than three RDs. There was no significant difference in the self-estimation of the level of knowledge of RDs (*p* = 0.196). Knowledge was not described as “very good” or “inadequate”. The main hypothesis was confirmed: dentists at university hospitals had a greater level of knowledge than dentists outside university hospitals.

One study with 165 participants determined the awareness of Polish physicians for RDs [12]. This study targeted physicians who had attended specialization courses at the Poznan University of Medical Sciences. A total of 70.2% (*n* = 165) of the 235 physicians who had been contacted completed the questionnaire. A total of 55.2% (*n* = 91) of the physicians stated that they had been practicing their profession for up to 5 years, and 23% (*n* = 38) had 6–10 years of professional experience, with 10.9% (*n* = 18) having been in practice for longer than 15 years. Although 75.2% (*n* = 124) of the participants stated that they had encountered one person suffering from an RD in the course of their clinical activities, 22.4% (*n* = 37) had not. A total of 11.5% (*n* = 19) of the physicians also stated that a member of their family was suffering from an RD. Only 18.2% (*n* = 30) of the physicians knew the frequency and prevalence of RD, and 20.6% (*n* = 34) correctly estimated the number of RDs. A total of 53.9% (*n* = 89) were aware that children were primarily affected by RDs. A total of 16.4% (*n* = 27) of all physicians interviewed knew that 300–350,000,000 people suffered from RDs on a global scale. The participants were also presented with a list of 28 diseases (including 18 RDs) and were asked to select the diseases that they believed were rare. Niemann-Pick disease (79.4%, *n* = 131), Pompe disease (76.4%, *n* = 126), and Gaucher disease (73.9%, *n* = 122) were the most common. However, fewer participants identified generally known RDs, such as Duchenne muscular dystrophy (50.3%, *n* = 83), neurofibromatosis (37%, *n* = 61), Fragile X Syndrome (52.1%, *n* = 86) or phenylketonuria (49.1%, *n* = 81). Even fewer physicians were aware of cystic fibrosis (40%, *n* = 66), achondroplasia (33.9%, *n* = 56), acromegaly (20%, *n* = 33) and sickle cell anemia (8.9%, *n* = 13). Münchhausen syndrome (25.4%, *n* = 42), fibromyalgia (15.1%, *n* = 25) and halitosis (14.5%, *n* = 24) were the most frequent diseases confused with RDs. When asked whether RDs present a serious public health problem, 25.5% (*n* = 42) answered “definitely yes”, another 57.6% (*n* = 95) answered “yes”, 10.9% (*n* = 18) answered “no” and 1.2% (*n* = 2) responded ”definitely not”. Eight participants (4.8%) were unable to answer this question. A very conspicuous aspect was that 35.2% (*n* = 58) described their knowledge of RD as “very poor”. A total of 59.4% (*n* = 98) described it as inadequate, and only one physician (0.6%) estimated his knowledge of this subject as being “very good”. Most of the physicians “do not tend to” (46.7%, *n* = 77) or “definitely do not” (46.7%, *n* = 77) feel prepared to care for a patient suffering from an RD, which explained why 83% (*n* = 137) of the participants would like to improve their knowledge of RD. A total of 69.7% (*n* = 115) of 165 participants attended further medical education on this subject. When asked whether a compulsory training course on RD should be offered during medical training, 23% (*n* = 38) replied “definitely yes” and 53.3% (*n* = 88) replied “tendency toward yes”. It was the opinion of 55.7% (*n* = 92) of participants that geneticists should receive special training in RDs, and an additional 53.9% tended toward specialized training for pediatricians.

Walkowiak and Domaradzki (2021) offered possible solution approaches [12]. The awareness of and interest in RD could be aroused during medical training so that the curricula at universities in Poland should include compulsory tutoring programs on this subject. Because the instructing of future physicians on each of the RDs would be impossible, the students should primarily familiarize themselves with the most frequent RDs so that they can diagnose them faster in clinical practice. The awareness should also be created and enhanced by providing instruction to future physicians on the etiology, diagnosis and therapy of these diseases and the implications that an RD has on the patients and their families. Subject-specific literature and eLearning platforms that include tutorials, webinars and PowerPoint presentations could be helpful.

The same research group performed another study of nursing, physiotherapy and medical students at the PUMS (Poznan University of Medical Sciences, Poznań, Poland) [13]. The survey was comprised of 28 questions: 22 related to the knowledge and attitudes on RD, and 6 questions related to demographic data. A total of 654 of the 862 contacted students completely answered the survey (75.9%). A total of 94.2% (113/120) of the contacted student nurses, 79% (173/219) of the trainee physiotherapists and 70.4% (368/523) of the medical students participated in the survey. A total of 98% (*n* = 643) of participants stated that they had heard the term “rare disease”. Only 8.4% (*n* = 62) were aware that an RD affected one in 2000 people. A total of 10.3% (*n* = 71) of the students answered this question correctly. Approximately one-third of all participants (33.5%, *n* = 204) were aware that children especially suffered from RDs. Only 10% (*n* = 67) correctly estimated the number of people on a global scale that suffered from an RD. In contrast, most of the participants were aware that RDs were most frequently of genetic origin (72%, *n* = 513). The students selected the diseases from a list of 28 diseases (including 18 RDs) that they classified as rare. The RDs that were most frequently recognized by the medical and nursing students were Morbus Pompe (medical students: 72.8%, (*n* = 268); nursing students: 51.3% (*n* = 58)), Gaucher disease (69%, *n* = 254 and 49.6%, *n* = 56) and Niemann-Pick disease (65.8%, *n* = 242 and 49.6%, *n* = 56), and the physiotherapy students selected Morbus Pompe (54.3%, *n* = 94), Niemann-Pick (49.7%, *n* = 86) and Fragile X syndrome (47.4%, *n* = 82). Within all three groups, Münchhausen syndrome was confused most frequently with RD. A total of 50.5% (*n* = 186) of the medical students, 49.1% (*n* = 85) of the physiotherapy students and 52.2% (*n* = 59) of the nursing students classified this disease as an RD. When answering the question on the name of the European website that provides information on RD, only 20.6% (*n* = 76) of the medical students, 11% (*n* = 19) of the physiotherapy students and 0.9% (*n* = 1) of the nursing students gave the correct answer of “Orphanet”. A total of 38.9% (*n* = 143) of the medical students deemed their knowledge of RD to be “very poor”, with an additional 56.2% (*n* = 207) stating that it was “inadequate”. A total of 38.7% (*n* = 67) and 56.1% (*n* = 97) of the physiotherapy students, respectively, also stated this, and 54% (*n* = 61) and 40.7% (*n* = 46) of the nursing students, respectively. Ninety-five percent of the students in each of the groups estimated their knowledge of RD as “inadequate” or “very poor”. Almost 92% (*n* = 338) of the medical students and 84% of the physiotherapy (*n* = 145) and nursing (*n* = 95) students did not feel that they were able to care for patients suffering from a RD. However, 11% (*n* = 12) of the physiotherapy students were prepared to treat RD patients. When asked whether the participants wished to improve their knowledge of RD, 73.9% (*n* = 272) of the medical students, 85% (*n* = 147) of the physiotherapy students and 83.2% (*n* = 94) of the nursing students replied “yes”.

The knowledge of Israeli dentists/physicians of RD was also investigated [16]. General dentists, specialists in oral medicine or oral and maxillofacial surgery, orthodontists, periodontists, pediatric dentists, endodontists and prosthodontists participated in a survey. The total number of participants was 309. Most participants were male (*n* = 192, 62.1%) and general dentists (*n* = 211, 68.3%). They received their dental education and licensure in Israel (*n* = 225, 72.8%) and worked exclusively in private practice (*n* = 152, 49.2%) or private and public dental practices (*n* = 109, 35.2%). A total of 43.6% (*n* = 92) of general dentists knew the definition of an RD, and 12.8% (*n* = 27) did not know a specific definition. Among specialists in oral medicine or OMFS, 57.1% (*n* = 12) knew the correct answer. Only 10% (*n* = 21) of general dentists; 19% (*n* = 4) of specialists in oral medicine or oral and maxillofacial surgery; 13.6% (*n* = 6) from the group of orthodontists, periodontists, pediatric dentists, and endodontists; and 18.2% (*n* = 6) from the field of prosthodontics knew that 15% of RDs manifest in the orofacial region. The survey further asked whether the participant had ever treated or seen a patient with an RD. For most respondents, this question was answered “yes”. A total of 70.1% (*n* = 148) of general dentists; 95.2% (*n* = 20) of specialists in oral medicine or OMFS; 88.6% (*n* = 39) of orthodontists, periodontists, pediatric dentists, and endodontists; and 87.9% (*n* = 29) from the field of prosthodontics answered “yes”. A total of 14 different RDs with possible manifestations in the orofacial region were presented. Using epidermolysis bullosa as an example, 95.2% (*n* = 20) of specialists in oral medicine or OMFS recognized this condition, but only 48.3% (*n* = 102) of general dentists did. A large proportion of participants were aware that they needed more information on RDs in their daily practice (33.6% (*n* = 71) of general dentists, 61.9% (*n* = 13) of specialists in oral medicine or OMFS; 40.9% (*n* = 18) from the group of orthodontists, periodontists, pediatric dentists, and endodontists; and 42.4% (*n* = 14) from the specialty of dental prosthodontics) [10]. As a source of information, 95.2% (*n* = 20) of the specialists in oral medicine and OMFS used journals, and only 50.2% (*n* = 106) of general dentists used this source of information. Predominantly, dentists were asked for advice (66.8% (*n* = 141)). A total of 77.3–90.5% of the participants thought that they needed to increase their knowledge of different treatment modalities. Finally, 66.7–81.8% considered knowledge of RD with orofacial manifestations as a crucial factor in the differential diagnosis in their clinical activities.

Table 10 provides an overview of the studies discussed. The present study highlights an insufficient (dental) medical education in the field of RDs and the importance of continuous postgraduate (dental) medical education. In summary, there was demonstrable interest among the respondents, and student and postgraduate training programs were offered very sparsely.

With regard to the conceptualization of the present study in comparison with the publications listed in Table 10, it can be noted that “only” comparisons between different professional groups or institutions have been carried out so far using the same questionnaires. However, since these are difficult to compare, the present work served to classify the different institutions according to their structures and to develop questionnaires adapted to each of them in order to highlight the different needs and conditions. In comparison to the other studies, our investigation aims to find out how the current management of patients with a rare disease with orofacial involvement is perceived and what incentives (e.g., telematic infrastructure) need to be created in order to be able to deal with these patients more intensively. For example, our questionnaire avoids examinations of respondents about rare diseases. We want to find out how the cooperation with other colleagues works for patients with rare diseases, and we also address the financial remuneration for the treatments in our questionnaire.

At the time of medical licensure, a prospective physician knows an average of 2000 to 2500 predominantly common symptoms and diseases, especially from the medical specialties of internal medicine, neurology and surgery [17]. Although the time to make a correct diagnosis has shortened over the past several years, it still takes far too long for patients to obtain clarity about their condition. On average, eight different physicians are consulted, and it takes an average of five years from clinical manifestation to medical diagnosis. Many people with RD are (initially) misdiagnosed and/or mistreated or remain undiagnosed and untreated despite high morbidity and mortality [18]. Providing the necessary interdisciplinary medical care at established centers is often not possible. The medical data of patients who are generally cared for in a decentralized manner are rarely pooled. However, efforts are being made to improve this situation. A new licensing regulation for dentists in Germany was enacted in 2021. For the first time, this regulation stipulates that the teaching of knowledge in the fields of geriatrics, patients with disabilities and rare diseases must be mandatory [19]. However, for the time being, this approach is only beneficial for students. An alternative for already practicing physicians and dentists is to read scientific articles. A crucial problem in this context is that scientific journals are less inclined to publish articles on RD as this often results in fewer citations [20].

### Limitations of the Study

The results of this study are exclusively based on the replies provided by the study participants. The self-estimations showed how they perceived their individual level of knowledge. It is not possible to show why they gave this answer and whether it correlates with the treatment situation for their patients. It is also the initial result of a national survey on the level of knowledge of RDs among dentists and OMFS. We can only compare the knowledge of physicians and their handling of the diagnostics and therapy of RD.

A bias must be assumed because presumably only participants who were interested in the topic completed the questionnaire.

## 5. Conclusions

Over a period of a decade (2011 to 2021), the available research shows that the training and thus the teaching of knowledge about rare diseases has not improved among an overwhelming number of doctors and dentists. Most of the participants believed that knowledge of RDs is very important for daily dental practice. General dentists and dental specialists primarily did not have any knowledge or only slight knowledge of RD. These self-estimations showed that all of the participants estimated their knowledge as very good or inadequate, with the tendency in the direction of inadequate knowledge. Knowledge of dentists and physicians on RDs appeared inadequate for suitable screening and therapy. This article suggests that more extensive knowledge is required in medicine and dentistry for people with RDs to receive a short path between the manifestation of the symptoms to diagnosis and treatment.

## Figures and Tables

**Table 1 medicina-58-01114-t001:** Questions on the person, work environment and qualification (Section I).

Question		*n* (%)	Yes	No	*n* (%)
1.	Total	309 (100%)	279 (90.3%)	30 (9.7%)	279 (100%)
	Salaried employees	90 (29.1%)	76 (84.4%)	14 (15.6%)	76 (27.2%)
	Self-employed	219 (70.9%)	203 (92.7%)	16 (7.3%)	203 (72.8%)
2.	Female	165 (53.4%)	146 (88.5%)	19 (11.5%)	146 (52.3%)
	Male	141 (45.6%)	130 (92.2%)	11 (7.8%)	130 (46.6%)
	Diverse	3 (1.0%)	3 (100%)	0 (0%)	3 (1.1%)
3.	Rural	136 (44.2%)	124 (91.2%)	12 (8.8%)	124 (44.4%)
	Urban	172 (55.8%)	155 (90.1%)	17 (9.9%)	155 (55.6%)
4.	License and private	299 (97.1%)	271 (90.6%)	28 (9.4%)	271 (97.1%)
	Only private	9 (2.9%)	8 (88.9%)	1 (11.1%)	8 (2.9%)
5.	Single dentist	128 (41.6%)	118 (92.2%)	10 (7.8%)	118 (42.3%)
	Colleagues in the Practice	180 (58.4%)	161 (89.4%)	19 (10.6%)	161 (57.7%)
6.	Less than 11 years	104 (33.9%)	91 (87.5%)	13 (12.5%)	91 (32.6%)
	Between 11 and 20 years	64 (20.8%)	58 (90.6%)	6 (9.4%)	58 (20.8%)
	Between 21 and 30 years	78 (25.4%)	71 (91.0%)	7 (9.0%)	71 (25.4%)
	Between 31 and 40 years	49 (16.0%)	47 (95.9%)	2 (4.1%)	47 (16.8%)
	Longer than 40 years	12 (3.9%)	12 (100%)	0 (0%)	12 (4.3%)
7a.	Oral surgery	32 (10.4%)	30 (93.8%)	2 (6.2%)	30 (10.8%)
	Orthodontics	9 (2.9%)	8 (88.9%)	1 (11.1%)	8 (2.9%)
	Periodontology	7 (2.3%)	7 (100%)	0 (0%)	7 (2.5%)
	Public health service	2 (0.6%)	1 (50.0%)	1 (50.0%)	1 (0.4%)
	No	258 (83.8%)	233 (90.3%)	25 (9.7%)	233 (83.5%)
7b.	Yes	11 (3.6%)	11 (100%)	0 (0%)	11 (3.9%)
	No	297 (96.4%)	268 (90.2%)	29 (9.8%)	268 (96.1%)
8.	Yes	215 (70.3%)	198 (92.1%)	17 (7.9%)	198 (71.0%)
	No	91 (29.7%)	81 (89.0%)	10 (11.0%)	81 (29.0%)
9.	Frequently	133 (61.9%)	122 (91.7%)	11 (8.3%)	122 (61.6%)
	Seldomly	82 (38.1%)	76 (92.7%)	6 (7.3%)	76 (38.4%)
10.	Pointless in my opinion	13 (4.2%)	13 (100%)	0 (0%)	13 (4.7%)
	Never	124 (40.4%)	111 (89.5%)	13 (10.5%)	111 (39.8%)
	Occassionally	116 (37.8%)	105 (90.5%)	11 (9.5%)	105 (37.6%)
	Regularly	54 (17.6%)	50 (92.6%)	4 (7.4%)	50 (17.9%)

1. Are you self-employed or a salaried employee in your practice? 2. Are you female or male? 3. Do you work in a rural or an urban area? 4. Do you have a health insurance license or has your employment been approved by the admissions committee or the health insurance dentists’ association and/or do you exclusively provide private dental treatment? 5. Do you work together with dental colleagues in your practice, or are you a dentist practising alone? 6. How long have you been providing outpatient treatment as a dentist? 7a. Are you a specialized dentist/training assistant for oral surgery, orthodontics, periodontology or the public health service? 7b. Are you a specialized dentist/training assistant for oral and maxillofacial surgery? 8. Do you use a telematic infrastructure in your practice? 9. If so, how frequently do you use your telematic infrastructure? 10. Do you participate in a quality circle?

**Table 2 medicina-58-01114-t002:** Age of participants (years).

Sections II–IV	*n*	Mw	Sd	Median	Min–Max
answered	279	47.4	12.8	48.0	24.0–100.0
not answered	27	41.4	14.6	43.0	1.0–66.0
total	306	46.9	13.0	47.0	1.0–100.0

**Table 3 medicina-58-01114-t003:** Questions on professional cooperation in the field of rare diseases with orofacial participation (Section II).

Question		*n* (%)
	Non-single practice	101 (36.2%)
	Single practice	178 (63.8%)
1a.	Never	7 (6.9%)
	Seldom	62 (61.4%)
	Regularly	32 (31.7%)
1b.	Rare disease diagnosed; co-treatment necessary	40 (39.6%)
	Obtaining a second opinion	83 (82.2%)
	Wish of the patient	15 (14.9%)
1c.	Never	8 (7.9%)
	Seldom	60 (59.4%)
	Regularly	33 (32.7%)
1d.	Rare disease	45 (44.6%)
	Obtaining a second opinion	84 (83.2%)
	Wish of the patient	15 (14.9%)
2a.	Never	14 (7.9%)
	Seldom	86 (48.3%)
	Regularly	78 (43.8%)
2b.	0–25%	95 (53.7%)
	26–50%	11 (6.2%)
	51–75%	20 (11.3%)
	76–100%	51 (28.8%)
2c.	0%	40 (22.7%)
	Max. 5%	81 (46.0%)
	Max. 10%	18 (10.2%)
	More than 10%	37 (21.0%)

1a. Have patients suffering from rare diseases with orofacial participation presented themselves, resulting in you consulting with your colleagues in the practice in the sense of a dental consultation? 1b. Can you describe the reason for the consultation (multiple answers are possible). 1c. Do you have cases where patients suffer from rare diseases that are connected to dental treatment and you then consult with your colleagues in the practice in connection with this in the sense of a medical consultation? 1d. Can you describe the reason for the consultation (multiple answers are possible) 2a. Do you consult with external dentists/oral and maxillofacial surgeons regarding the treatment of patients suffering from rare diseases? 2b. With what frequency do you refer patients suffering from rare diseases to specialist dentists or specialist physicians? 2c. With what frequency do you refer patients suffering from rare diseases to university dental hospitals/clinics for oral and maxillofacial surgery?

**Table 4 medicina-58-01114-t004:** Questions on the use of a register for rare diseases with an orofacial manifestation (Section III).

Question		*n* (%)
1.	Yes	236 (87.7%)
	No	5 (1.9%)
	Uncertain	28 (10.4%)
2.	Yes	237 (88.1%)
	No	5 (1.9%)
	Uncertain	27 (10.0%)
3.	Medline, PubMed	123 (45.4%)
	Textbooks	219 (80.8%)
	other sources	97 (35.8%)
	None	13 (4.8%)
4.	Orphanet	27 (10.0%)
	OMIM	5 (1.8%)
	Se-atlas	4 (1.5%)
	ROMSE	12 (4.4%)
	Other databases	16 (5.9%)
	None	220 (81.2%)
5.	Congresses/symposiums	163 (60.1%)
	Interdisciplinary quality circle	115 (42.4%)
	Advanced telemedical training	149 (55.0%)
	Other advanced training	23 (8.5%)

1. Do you deem the creation of such a register as being helpful for the diagnosis and treatment of patients suffering from rare diseases? 2. Do you see any benefits from such a register with regard to communication with medical colleagues? 3. Which sources of information do you use in your practice for the diagnosis and treatment of patients suffering from rare diseases? 4. Which databases do you use in your practice for the diagnosis and treatment of patients suffering from rare diseases? 5. What forms of advanced training do you believe are appropriate for improving your knowledge of rare diseases?

**Table 5 medicina-58-01114-t005:** Questions on professional exchange in the field of rare diseases with orofacial participation (Section IV).

Question		*n* (%)
1.	Yes	206 (79.5%)
	No	13 (5.0%)
	Uncertain	40 (15.4%)
2.	Yes	14 (5.4%)
	No	237 (91.5%)
	Uncertain	8 (3.1%)
3.	Yes	11 (4.2%)
	No	236 (91.1%)
	Uncertain	12 (4.6%)
4.	Yes	106 (41.1%)
	No	82 (31.8%)
	Uncertain	70 (27.1%)
5.	Yes	106 (41.1%)
	No	61 (23.6%)
	Uncertain	91 (35.3%)
6.	Yes	189 (73.0%)
	No	17 (6.6%)
	Uncertain	53 (20.5%)
7.	Yes	155 (60.1%)
	No	33 (12.8%)
	Do not know	70 (27.1%)
If yes:	With numerous meetings	63 (28.0%)
	By presenting individual cases	205 (91.1%)
8.	Yes	54 (20.8%)
(Column BD)	No	205 (79.2%)
If ‘yes’ under 8:9.	Yes	37 (68.5%)
(Column BJ)	No	17 (31.5%)
9.	Yes	207 (79.9%)
(Column BF)	No	52 (20.1%)
If ‘yes’ under 8:9.	Yes	18 (33.3%)
(Column BL)	No	36 (66.7%)
10.	Yes	209 (86.7%)
	No	13 (5.4%)
	No opinion	19 (7.9%)

1. Would you like to cooperate with colleagues to a greater extent and benefit from the special knowledge that other colleagues have so that you can treat patients suffering from a rare disease? 2. Would you be afraid of losing patients if you were to refer patients suffering from rare diseases to a colleague in order to clarify a dental medicine aspect? 3. Are you afraid that there could be doubts regarding your skills, if you should have to refer patients suffering from rare diseases to another colleague? 4. Is it your opinion that the dental treatment of patients suffering from rare diseases should be supported with payment incentives? 5. Do you believe that a telematic infrastructure for communication among dentists and physicians is appropriate for making the dental diagnosis and treatment of patients suffering from rare diseases easier? 6. Would you avail yourself of an offer made by a telemedical center that is connected to a university dental hospital to clarify specialization-related aspects in patients suffering from rare diseases? 7. Would you like to see the knowledge regarding patients suffering from rare diseases being intensified in quality circles that regularly take place? 8. Do you know any centers for rare diseases? 9. If “yes” under 8: Have you already cooperated with such a center? 10. Do you see the subject of rare diseases as being clinically relevant?

**Table 6 medicina-58-01114-t006:** Questions on the department.

Question	Question		*n* (%)
		Total	18 (100%)
1.		MHH Hanover	1 (5.6%)
		Münster	2 (11.1%)
		RWTH Aachen	2 (11.1%)
		Tübingen	1 (5.6%)
		Frankfurt am Main	1 (5.6%)
		Witten/Herdecke	1 (5.6%)
		No information	10 (55.6%)
2.		Chair and management	2 (11.1%)
		Senior Physician	5 (27.8%)
		Other	11 (61.1%)
3.		1–3	8 (44.4%)
		>3	10 (55.6%)
4.		1–5	6 (33.3%)
		6–10	2 (11.1%)
		11–15	7 (38.9%)
		>15	3 (16.7%)
5.		All clinicians, depending on free capacities	12 (66.7%)
		Only certain clinicians	6 (33.3%)
		Number of clinicians:	
		1	2 (33.3%)
		2	2 (33.3%)
		2–3	1 (16.7%)
		3	1 (16.7%)
6.		Yes	4 (22.2%)
		No	14 (77.8%)
	If yes:	Regular advanced training	2 (50.0%)
		Other further training	3 (75.0%)
		Own clinical experience	3 (75.0%)
7.		Yes	2 (11.1%)
		No	16 (88.9%)
	If yes:	Within the department	2 (100%)
		Interdisciplinary	2 (100%)
		As a ‘board’	2 (100%)
		Separate treatment times for patients suffering from rare diseases?	2 (100%)

1. Which university do you work at? 2. What is your function within your department? 3. How many senior physicians does your department have? 4. How many scientific employees (dentists/oral and maxillofacial surgeons) work in your department? 5. How many clinicians in your department treat patients suffering from rare diseases? 6. Have these clinicians received training in the field of rare diseases to a special extent? 7. Does your department offer surgery for rare diseases?

**Table 7 medicina-58-01114-t007:** Questions on the patients.

Question		*n* (%)
8.	1–10	6 (37.5%)
	11–20	2 (12.5%)
	>20	8 (50.0%)
9.	Referrals within the university	11 (68.8%)
	External specialist departments	8 (50.0%)
	External practices	12 (75.0%)
	From abroad	2 (12.5%)
10.	Only intraoral	8 (50.0%)
	Intraoral and extraoral	8 (50.0%)

8. How many patients suffering from rare diseases does your department treat on an annual basis? 9. Where does this group of patients come from? 10. Which orofacial symptoms of rare diseases affecting the oral and maxillofacial area have been treated in your department?

**Table 8 medicina-58-01114-t008:** Questions on research in the rare diseases field.

Question		*n* (%)
11.	Yes	2 (13.3%)
	No	13 (86.7%)
If yes:		
11.1	0	1 (50.0%)
	20	1 (50.0%)
11.2	Yes	1 (50.0%)
	No	1 (50.0%)

11. Is research on rare diseases carried out in your department? 11.1. How many scientific studies have been published on the subject of “rare diseases” by your department during the past three years? 11.2. Has your department been represented by a specialist lecture on the subject of rare diseases that have been held at national/international congresses?

**Table 9 medicina-58-01114-t009:** Questions on lectures.

Question			*n* (%)
12.		Yes	4 (36.4%)
		No	7 (63.6%)
13.		Yes	2 (18.2%)
		No	3 (27.3%)
		Unknown	6 (54.5%)
14.		Yes	5 (45.5%)
		No	6 (54.5%)
14.1	If yes, in what form:	Sitting in/clinical traineeship	2 (40.0%)
		Internship	3 (60.0%)

12. Do you offer a lecture (series of lectures) on the subject of “Rare Diseases and Dental, Oral and Orthodontic Medicine/Maxillofacial Surgery” to students at your university? 13. Do different faculty/other faculty members offer lectures on the subject of “random diseases” at your university? (yes: faculty) 14. Are students able to participate in the treatment of the group of patients suffering from rare diseases in your faculty? 14.1. What form does this participation have?

**Table 10 medicina-58-01114-t010:** Overview of published studies on awareness and knowledge of RDs.

Authors	Target Group	Number of Participants	Type of Survey	Aim	Survey Structure	Conclusion
Miteva et al., 2011 [14]	General practitioners (GP)	1002	Phone interview 17 questions	Evaluation of the knowledge and awareness of RD	Demographics Definition and frequency of RD Specific knowledge related to RD Treatment of RD Interests in RD	“A campaign for increasing the awareness of GPs about rare diseases (prevention, diagnosis, treatment and rehabilitation) is needed.”
Vandeborne et al. 2019 [11]	GPs, neurologists, endocrinologists, pediatricians, neuropediatricians, pediatric endocrinologists	295 GPs 25 specialists	Online survey	Identifying how information and education can be tailored to the needs and preferences of GPs in Belgium to raise awareness of RD and assist them in diagnosing patients with RD	Demographics Level of knowledge in relation to RD “Special awareness” in relation to RD Continuing education on the topic of RD Needs of practitioners in relation to RD	“(…), the academic medical education on rare diseases should be revised.”
Kuhne et al., 2020 [15]	In private practice/university: Dentists Specialized dentists OMFS	267	Online survey	Survey of the level of knowledge about RD among dentists at university hospitals compared to dentists with different backgrounds (general dentists, specialized dentists, OMFS) in the chamber area of Westphalia-Lippe	Demographics Knowledge on special rare diseases Experiences with RD	“(…) the current knowledge of the dental profession regarding RDs does not seem to be sufficient for adequate early detection and therapy.”
Walkowiak and Domaradzki, 2021 [12]	GPs in training at the University of Poznan, Poland	165	Survey	Assessment of awareness of RD among GPs in Poland	Demographics Definition, etiology and estimated prevalence Education and knowledge related to RD	“(…), while this study has identified the gap in physicians’ knowledge on RDs, (…) their empowerment is a huge project that requires implementation of multilevel solutions.”
Domaradzki and Walkowiak, 2021 [13]	Nursing, physical therapy, and medical students in the final stage of their education	654	Survey	Assessment of RD awareness among nursing, physical therapy, and medical students in Poland	Demographics Knowledge related to RD Organizational issues related to health care in Poland Training in RD during studies	“(…), that there is a serious educational gap in future healthcare professionals about RDs.”“(…) that there is an urgent need to include RDs into the medical curricula.”
Mijiritsky et al., 2021 [16]	Dentists Specialized dentists OMFS	309	Online survey	Assessment of the levels of knowledge and associated factors of RD	Demographics Knowledge on special rare diseases Experiences with RD	“(…) revealed that the majority of dental practitioners in Israel claim that they require further information about these rare pathologies.”

## Data Availability

Not applicable.

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
