# Peer review of "Awareness and Knowledge of Rare Diseases in German Dentists, Dental Specialists and Oral and Maxillofacial Surgeons: A Country-Wide Survey"

_medicina, 2022, doi:10.3390/medicina58081114_

Round 1

Reviewer 1 Report

Advantages: The content of the manuscript contains a variety of information about awareness of rare diseases. The questionnaires are well organized and they are easy to answer for all of the questions. 

Comment 1: Since the articles was described about the rare disease, the author need to add the example of rare diseases, like top 5 rare diseases related to oral cavity. 

Comment 2: When describing the result of table 1., the important data should describe more clearly.

Comment 3: While writing the result, table 4&5 didn’t mentioned in the paragraph.

Comment 4: In discussion, the informations from the previous articles are too much compared to your current study that might confuse to the reader.

Comment 5: For table 10, the current study should also compare with published articles that can point out more about the Novelty of your manuscript.

Suggestion: However, the contents in the manuscript is truly reflect the current situation of handling the rare diseases. Mainly the results and discussion part should reorganize to present a clear description of the data.

Author Response

Advantages: The content of the manuscript contains a variety of information about awareness of rare diseases. The questionnaires are well organized and they are easy to answer for all of the questions. 

Comment 1: Since the articles was described about the rare disease, the author need to add the example of rare diseases, like top 5 rare diseases related to oral cavity. 

Answer: Thank you for this comment - we have now added a paragraph in the introduction (please see lines 58-69) describing various rare diseases which we think should be known to the readership at least by name including their orofacial manifestations. A "top 5" ranking is not possible because the total number of cases is too small.

Comment 2: When describing the result of table 1., the important data should describe more clearly.

Answer: Dear reviewer, please find the data described in lines 212-235, thank you very much!

Comment 3: While writing the result, table 4&5 didn’t mentioned in the paragraph.

Answer: Thank you for this important note, both tables are now mentioned in the paragraph.

Comment 4: In discussion, the informations from the previous articles are too much compared to your current study that might confuse to the reader.

Answer: Thank you for this comment. In preparing the manuscript, we considered that we would like to list the available data in great detail to provide the reader with as comprehensive an overview as possible, especially in light of the fact that very little data is available to date. To our knowledge, a presentation in this detailed form has not existed before. Nevertheless, we have shortened the discussion by a few points.

Comment 5: For table 10, the current study should also compare with published articles that can point out more about the Novelty of your manuscript.

Answer: We have now related our methodology to the concepts of the other studies and highlighted key differences and the novelty of the manuscript.

Suggestion: However, the contents in the manuscript is truly reflect the current situation of handling the rare diseases. Mainly the results and discussion part should reorganize to present a clear description of the data.

Answer: We sincerely hope that our revision will meet your expectations.

Reviewer 2 Report

the study is well written and executed, it adequately describes how it was carried out and precisely presents the obtained results. it is interesting to discover how rare diseases and treatments are poorly understood. maybe the discussion section is too prolix. this study could be a starting point for raising awareness on rare diseases, not only in Germany.

Author Response

the study is well written and executed, it adequately describes how it was carried out and precisely presents the obtained results. it is interesting to discover how rare diseases and treatments are poorly understood. maybe the discussion section is too prolix. this study could be a starting point for raising awareness on rare diseases, not only in Germany.

Answer: We are very happy about this positive comment. We have tried to shorten the discussion by a few points and hope that our revision will meet your expectations.

Reviewer 3 Report

The study is of interest and well conducted and gives the state of knowledge of the problem.

References are adequate for the study.

the main question addressed by the research was about the knowledge of rare diseases in Germany among dentistry operators. The paper is interesting, well written and clear to read and easily understood from readers.

Conclusions are consistent with the evidence and the arguments treated, and address the main question posed.

I suggested just a minor revision of English.

Author Response

The study is of interest and well conducted and gives the state of knowledge of the problem.

References are adequate for the study.

the main question addressed by the research was about the knowledge of rare diseases in Germany among dentistry operators. The paper is interesting, well written and clear to read and easily understood from readers. 

Conclusions are consistent with the evidence and the arguments treated, and address the main question posed.

I suggested just a minor revision of English.

Answer: Thank you for the positive review! We have had the manuscript proofread and corrected by American Journal Experts Publishing. Please find the certificate as a supplementary in the upload section.